# Voluntary Reduction of Social Interaction during the COVID-19 Pandemic in Taiwan: Related Factors and Association with Perceived Social Support

**DOI:** 10.3390/ijerph17218039

**Published:** 2020-10-31

**Authors:** Wei-Po Chou, Peng-Wei Wang, Shiou-Lan Chen, Yu-Ping Chang, Chia-Fen Wu, Wei-Hsin Lu, Cheng-Fang Yen

**Affiliations:** 1Graduate Institute of Medicine, College of Medicine, Kaohsiung Medical University, Kaohsiung 80708, Taiwan; webber1007@gmail.com (W.-P.C.); wistar.huang@gmail.com (P.-W.W.); shioulan@kmu.edu.tw (S.-L.C.); pino3015@hotmail.com (C.-F.W.); 2Department of Psychiatry, Kaohsiung Medical University Hospital, Kaohsiung 80708, Taiwan; 3School of Nursing, The State University of New York, University at Buffalo, New York, NY 14214-3079, USA; yc73@buffalo.edu; 4Department of Psychiatry, Ditmanson Medical Foundation Chia-Yi Christian Hospital, Chiayi City 60002, Taiwan

**Keywords:** COVID-19, social support, social interaction, health belief

## Abstract

This study aimed to determine the proportion of individuals who voluntarily reduced interaction with their family members, friends, and colleagues or classmates to avoid coronavirus disease 2019 (COVID-19) infection and the associations of reduced social interaction with perceived social support during the COVID-19 pandemic in Taiwan. Moreover, the related factors of voluntary reduction of social interaction were examined. We recruited participants via a Facebook advertisement. We determined the reduced social interaction, perceived social support, cognitive and affective constructs of health belief and demographic characteristics among 1954 respondents (1305 women and 649 men; mean age: 37.9 years with standard deviation 10.8 years). In total, 38.1% of respondents voluntarily reduced their social interaction with friends to avoid COVID-19 infection, 36.1% voluntarily reduced their interaction with colleagues or classmates, and 11.1% voluntarily reduced interaction with family members. Respondents who voluntarily reduced interaction with other people reported lower perceived social support than those who did not voluntarily reduce interaction. Respondents who were older and had a higher level of worry regarding contracting COVID-19 were more likely to voluntarily reduce interaction with family members, friends, and colleagues or classmates to avoid COVID-19 infection than respondents who were younger and had a lower level of worry regarding contracting COVID-19, respectively. The present study revealed that despite strict social distancing measures not being implemented in Taiwan, more than one-third of respondents voluntarily reduced their interaction with friends and colleagues or classmates. The general public should be encouraged to maintain social contacts through appropriately distanced in-person visits and telecommunication.

## 1. Introduction

### 1.1. Reduction of Social Interaction and Social Support during the Coronavirus Disease 2019 Pandemic

The coronavirus disease 2019 (COVID-19) has had an enormous impact on numerous dimensions of human lives, including health [1,2,3], work [4], education [5], leisure activities [6], and economic action [7]. The change in social interaction is an especially obvious and far-reaching effect of the COVID-19 pandemic. A study in 25 European countries revealed that higher perceived sociability among respondents was a predictor of higher COVID-19 mortality in their countries [8]. COVID-19 is highly contagious [9], which has forced governments worldwide to implement policies to reduce the spread of the infection, such as quarantine, lockdown, closure of nonessential stores and educational institutions, and border closure [10]. However, fewer regulation policies are implemented in the regions where the effects of the COVID-19 pandemic are less severe. In situations where social distancing mostly remains voluntary, the public may choose to reduce the frequency of social interaction during the COVID-19 pandemic.

Studies have revealed that although social distancing mitigates the spread of respiratory infectious diseases [11], it may increase the possibility of social isolation, which increases the risk of morbidity and mortality [12]. Research has indicated that social distancing and isolation from social networks were associated with negative mental health trajectories during the COVID-19 pandemic [13,14]. Reduced social support has been proposed as a mediator of the association of social distancing with negative mental health outcomes. Social support refers to the access of an individual to support through social ties to other individuals, groups, and the larger community [15]. Studies conducted during the COVID-19 pandemic have revealed that a higher level of perceived social support was associated with fewer COVID-19-related worries in the general public [16] and fewer acute stress disorder symptoms among college students [17]. A study in medical staff during the COVID-19 pandemic revealed that a higher level of social support was positively associated with self-efficacy and higher sleep quality and negatively associated with the degree of anxiety and stress [18,19]. Social support was also crucial in the retention in care of older people with human immunodeficiency virus infection and substance use disorders during the COVID-19 pandemic [20]. These results support that perceived social support is a critical protector of the mental health of individuals during the COVID-19 pandemic. However, the effects of voluntarily reducing the frequency of social interaction on the level of perceived social support in the general public during the COVID-19 pandemic warrant further study. The results of the study may assist in the development of strategies that would provide the necessary social connections [21].

### 1.2. Factors Related to the Reduction of Social Interaction

The Health Belief Model (HBM) indicates that worry regarding falling ill is a health belief construct that predicts individuals’ engagement in health-related behaviors [22,23]. A study reported that individuals who perceived a higher risk of contracting COVID-19 were more likely to adopt social distancing behaviors [24]. Worry regarding contracting COVID-19 may have similar effects on individuals’ voluntary social interaction with their family, friends, and colleagues or classmates.

In contrast to the specific worry of contracting COVID-19, general anxiety is persistent, as well as excessive worry regarding numerous things, such as disaster, money, health, family, and work. During the COVID-19 pandemic, general anxiety may develop in both healthy individuals and people with preexisting mental health conditions [25]. The relationship between general anxiety and the reduction of social interaction during the COVID pandemic remains unclear. General anxiety may compromise individuals’ working memory [26], which is essential for the awareness of benefits over costs of social distancing [27]. General anxiety may also compromise the motivation to socially interact with other people [28]. The association between general anxiety and the voluntary reduction of social interaction warrants further study.

Research has revealed disparities in COVID-19 incidence, knowledge, and behavior among people with various demographic characteristics. A study in China revealed that individuals aged ≥65 years were more susceptible to COVID-19 infection [29]. A study in the United States reported that people aged <30 years were more likely to know someone who tested positive for COVID-19 than other age groups [30]. A study in the United States revealed that men possessed less COVID-19-related knowledge compared with women [30]. Furthermore, a study in Switzerland determined that women had worse mental health during the pandemic compared with men [14]. The associations of demographic characteristics with the voluntary reduction of social interaction in the general public warrant further study.

### 1.3. COVID-19 Pandemic in Taiwan

The number of COVID-19 cases in Taiwan is very low compared with numerous other countries or regions because of the early adoption of proactive containment and comprehensive contact tracing [31]. By 1 October 2020, 514 confirmed cases had been identified in Taiwan, among which 55 were domestic and 7 were fatal [32]. Taiwan experienced a major outbreak of 2002–2003 severe acute respiratory syndrome (SARS), with the third-highest number of SARS cases after China and Hong Kong [33]. The traumatic experience of the SARS epidemic increased the vigilance of Taiwanese people toward COVID-19. People in Taiwan may voluntarily adopt protective behaviors, including reducing social interaction, despite the absence of government-imposed social lockdown. The extent of voluntarily reduced social interaction during the COVID-19 pandemic and the factors related to the reduction of social interaction in Taiwan, where social distancing mostly remains voluntary, were investigated; the related results may contribute to the establishment of strategies to control the spread of respiratory infectious diseases in the future.

### 1.4. Aims of This Study

The present study had three aims. First, we examined the proportions of individuals who voluntarily reduced interaction with their family members, friends, and colleagues or classmates to avoid COVID-19 infection. Second, we examined the associations of reduced social interaction with perceived social support during the COVID-19 pandemic. Third, we examined the associations of demographic characteristics, worry regarding contracting COVID-19, and general anxiety with the reduction of social interaction during the COVID-19 pandemic.

## 2. Methods

### 2.1. Participants

The current investigation was based on dataset of the Survey of Health Behaviors During the COVID-19 Pandemic in Taiwan, which was comprehensively described elsewhere [34]. Briefly, a Facebook advertisement was deployed between 10 April and 23 April 2020. By 23 April 2020, 427 confirmed cases had been identified in Taiwan, of which six were fatal [32]. The inclusion criteria for participants were aged 20 years or older and residing in Taiwan. The Facebook advertisement included a headline, main text, pop-up banner, and link to the research questionnaire website. We designed the advertisement to appear in the Facebook users’ “news feeds,” which is a continually updated list of updates from advertisers and the user’s connections (such as friends and the Facebook groups that they have joined). Our advertisement only targeted users’ news feeds, as opposed to other Facebook advertising locations (e.g., the right column), because news feed advertisements are most effective in recruiting research participants [35]. We targeted the advertisement to Facebook users by location (Taiwan) and language (Chinese), where Facebook’s advertising algorithm determined which users to show our advertisement to. The survey administration software Google Forms was used for data collection (https://www.google.com/forms/about/). The Institutional Review Board of Kaohsiung Medical University Hospital (Kaohsiung City, Taiwan) approved this study (KMUHIRB-EXEMPT(I) 20200011) and ruled that informed consent was not required. At the end of the online questionnaire, participants were provided links to the COVID-19 sections of the websites of the Taiwan Centers for Disease Control, Kaohsiung Medical University Hospital, and Medical College of National Cheng Kung University (Tainan City, Taiwan), to access COVID-19-related information and knowledge.

### 2.2. Measures

#### 2.2.1. Voluntary Reduction of Social Interaction

The voluntary reduction of social interaction because of COVID-19 was measured with the following three-part question: “In the last week, did you voluntarily reduce the frequency of interaction with your (1) family members, (2) friends, and (3) colleagues or classmates?” The response options were as follows: 1, “No”; 2, “Yes, but not to avoid COVID-19 infection”; and 3, “Yes, to avoid COVID-19 infection.” Responses 1 and 2 were classified as not voluntarily reducing social interaction during the COVID-19 pandemic, and response 3 was classified as voluntarily reducing social interaction.

#### 2.2.2. Perceived Social Support

Perceived social support was measured with the following three-part question: “In the last week, were you satisfied with the support from your (1) family member, (2) friends, and (3) colleagues or classmates?” The response options were as follows: 0, “Not satisfied at all”; 1, “Mildly unsatisfied”; 2, “Fair”; 3, “Satisfied”; and 4, “Very satisfied.”

#### 2.2.3. Worry about Contracting COVID-19

A five-item questionnaire was used to measure the level of worry regarding contracting COVID-19 [36]. The five items assessed the respondents’ level of worry if they developed flu-like symptoms, level of worry regarding contracting COVID-19, level of worry toward COVID-19, likelihood of contracting COVID-19, and chances of contracting COVID-19 compared with others outside their family. The questions, response scales, and scoring are listed in Appendix A. The scores of the five questions were summed. The Cronbach’s α was 0.759. The median total score of 2.95 was used as the cutoff, and respondents with total scores of ≤2.95 and >2.95 were classified as the low and high worry groups, respectively.

#### 2.2.4. General Anxiety

The 10-item state anxiety scale from the State-Trait Anxiety Inventory was used to measure respondents’ general anxiety levels over the previous week [36,37,38]. The questions, response scales, and scoring are listed in Appendix A. The scores of the 10 items were summed. The Cronbach’s α was 0.921. The median total score of 23 was used as the cutoff, and respondents with total scores ≤23 and >23 were classified as the low and high general anxiety groups, respectively.

#### 2.2.5. Demographic Variables

Participants’ sex (women vs. men), age, and education level (high school or below vs. college or above) were recorded. The median age (37 years old) was used as the cutoff, and respondents aged ≤37 and >37 years were classified as the younger and older groups, respectively.

### 2.3. Statistical Analysis

Data collected were input, processed and analyzed using the statistical software SPSS (version 22.0; SPSS Inc., Chicago, IL, USA). First, the percentages of respondents who voluntarily reduced the frequency of social interaction with their family members, friends, and colleagues or classmates to avoid COVID-19 infection were calculated. Second, multiple regression analysis was performed to examine the associations of voluntarily reduced social interaction with perceived social support, controlling for the effects of demographic characteristics. Finally, the associations of demographic characteristics, worry regarding contracting COVID-19, and general anxiety with the voluntary reduction of social interaction were examined using multivariate logistic regression with odds ratio and 95% confidence intervals. A two-tailed *p*-value of less than 0.05 was considered statistically significant.

## 3. Results

### 3.1. Proportion of Respondents Who Voluntarily Reduced Their Social Interaction

In total, the data of 1954 respondents (649 men and 1305 women) were analyzed, with 70 of the original 2024 respondents excluded due to missing data. The demographic characteristics, percentage of respondents who voluntarily reduced their social interaction, levels of perceived social support, levels of worry regarding contracting COVID-19, and general anxiety are presented in Table 1. The results indicated that 38.1% of respondents voluntarily reduced their social interaction with friends to avoid COVID-19 infection, 36.1% voluntarily reduced their interaction with colleagues or classmates, and 11.1% voluntarily reduced interaction with family members; 23.2% and 9.3% of respondents voluntarily reduced two and three categories of social interaction, respectively.

### 3.2. Association between the Voluntary Reduction of Social Interaction and Perceived Social Support

The relationship between the voluntary reduction of social interaction and perceived social support is presented in Figure 1 and Table 2. The results indicated that voluntary reduction of interaction with family members, friends, and colleagues or classmates were negatively associated with perceived support from family members, friends, and colleagues or classmates, respectively, after controlling for the effects of demographic characteristics.

### 3.3. Factors Related to the Voluntary Reduction of Social Interaction

The factors related to each type of voluntary reduced social interaction are presented in Table 3. The results indicated that respondents who were older and had a higher level of worry regarding contracting COVID-19 were more likely to voluntarily reduce interaction with family members, friends, and colleagues or classmates to avoid COVID-19 infection than respondents who were younger and had a lower level of worry regarding contracting COVID-19, respectively. Men were more likely to voluntarily reduce interaction with colleagues or classmates than women. Respondents with a low education level were less likely to voluntarily reduce interaction with friends compared with respondents with a higher education level. Respondents who had a higher level of general anxiety were more likely to voluntarily reduce interaction with family members than respondents who had a lower level of general anxiety. 

## 4. Discussion

### 4.1. Voluntary Reduction of Social Interaction and Its Association with Perceived Social Support

The present study revealed that 38.1% and 36.1% of respondents voluntarily reduced interaction with their friends and colleagues or classmates to avoid COVID-19 infection, respectively, whereas only 11.1% voluntarily reduced interaction with family members. A study in China revealed that the strict social distancing measures implemented reduced people’s daily contacts sevenfold to eightfold during the COVID-19 pandemic, with most interaction restricted to the household members [29]. Because face-to-face interaction and random encounters are minimized as a result of the social distancing measures, individuals likely focus on the relationships that are spatially close, most meaningful, or most established [14], such as those in the household [39]. The present study revealed that despite strict social distancing measures not being implemented in Taiwan, more than one-third of respondents voluntarily reduced their interaction with friends and colleagues or classmates. Moreover, the present study revealed that respondents who voluntarily reduced interaction with other people reported lower perceived social support than those who did not voluntarily reduce interaction. Therefore, voluntary reduction of social interaction appears to cause feelings of isolation and negatively affects perceived social support, whereas more social support may be required to cope with stress caused by the COVID-19 pandemic [40]. Maintaining social contact can improve overall mental health, enhance feelings of social connectedness, and reduce loneliness. Therefore, the general public should be encouraged to maintain social contact through appropriately distanced in-person visits, telephone calls, video calls, and e-mail [21]. Novel models of social support and crisis intervention that utilize telecommunication, such as social media applications on smartphones, could be used during global pandemics [41]. However, people that are unable to access or use smartphones or the Internet have fewer opportunities to access social resources [42].

### 4.2. Factors Related to Voluntary Reduction of Social Interaction

The present study determined that a higher level of worry regarding contracting COVID-19 was positively associated with the voluntary reduction of interaction with family members, friends, and colleagues or classmates because of COVID-19. The HBM [22,23], indicates that perceived susceptibility to and severity of infectious diseases were the cognitive constructs of health beliefs that contribute to the adoption of preventive actions [43]. However, people with excessive worry may implement extreme measures to cope with COVID-19; for example, extreme social isolation may obstruct access to the social support necessary to cope the pandemic. Delivering timely and adequate information regarding COVID-19 is essential to instill a reasonable range of concerns regarding COVID-19 in the general public [44].

The study results revealed that older respondents were more likely to voluntarily reduce interaction with family members, friends, and colleagues or classmates as a result of COVID-19 than younger respondents. A study published in March 2020 revealed that elderly people in China are susceptible to COVID-19 infection and prone to serious outcomes, such as acute respiratory distress syndrome and cytokine storm [45]. The results were widely reported in Taiwan, which might have led older people to be vigilant and reduce social interaction to avoid contracting COVID-19. Moreover, older people in Taiwan may have clearer memories of SARS, which drives their proactive response to the COVID-19 pandemic, such as voluntarily reducing social interaction. However, prolonged social isolation is significantly associated with higher risks of cognitive impairment and physical problems among elderly people [46]. Balancing the benefits and damages of social isolation during the pandemic among elderly people is a critical topic that should be addressed by governments and health professionals.

Male respondents were more likely to voluntarily reduce interaction with colleagues or classmates than female respondents. Respondents with higher levels of general anxiety were more likely to voluntarily reduce interaction with family members than respondents with lower levels of general anxiety. Respondents with lower education levels were less likely to voluntarily reduce interaction with friends compared with those with higher education levels. Research in China revealed that male sex (vs. female), age group of 16–29 years (vs. 30–49 years), and education level of bachelor’s degree or lower (vs. master degree and above) were significantly associated with lower knowledge of COVID-19 [47]. Research also revealed that women were more likely to display negative emotional effects as a result of the pandemic; however, women also generally rely on denser support networks, which may help them buffer the negative effects of the crisis [14]. The reasons underlying the various associations between general anxiety and voluntary reduction of interaction with different groups warrant further study.

### 4.3. Future Research

There are several lines of studies worth further examination based on the data of this study. First, the prediction of personality, coping strategies and psychopathology for voluntary reduction of social interactions during the pandemic warrants study. Second, the follow-up studies are needed to determine the outcomes of the individuals with voluntary reduction of social interactions. Third, what programs that governments and nongovernmental organizations can adopt to increase social support warrants further study. Fourth, cross-cultural comparisons may provide knowledge to understand cultural differences in changes of social relationship during the pandemic.

### 4.4. Limitations

The present study has some limitations. First, although recruiting participants using the Facebook advertisement can deliver large numbers of participants quickly [48], Facebook users may not be representative of the population. Second, the cross-sectional design of this study limited causal inference between voluntary reduction of social interaction and perceived social support. Third, there might be factors such as personality characteristics that were not examined in the present study accounting for both voluntary reduction of social interaction and decreased social support. Fourth, participants completed the online questionnaire anonymously. Hence, it is impossible to verify the identity of the respondents and their responses in the survey. It needs to be clarified by a face-to-face interview study with examining the test-retest reliability.

## 5. Conclusions

This Facebook-based online study on the general public found that 38.1%, 36.1% and 11.1% of respondents voluntarily reduced interaction with their friends, colleagues or classmates, and family members to avoid COVID-19 infection, respectively, despite strict social distancing measures not being implemented in Taiwan. Moreover, respondents who voluntarily reduced interaction with other people reported lower perceived social support than those who did not voluntarily reduce interaction. The general public should be encouraged to maintain social contact through appropriately distanced in-person visits and telecommunication. The present study identified that age, sex, educational level, worry regarding contracting COVID-19 and general anxiety were significantly associated with all or some aspects of reduced social interactions during the pandemic. The results indicated that the changes in social interaction during the COVID-19 pandemic is a complicated phenomenon involving demographic characteristics and health beliefs. These factors can be used to early identify individuals who reduced social interaction with other people and then perceived low social support.

## Figures and Tables

**Figure 1 ijerph-17-08039-f001:**
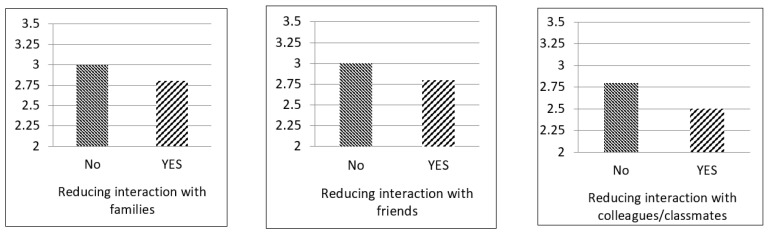
Levels of Perceived Social Support in the Participants with and Without Reduction of Social Interaction.

**Table 1 ijerph-17-08039-t001:** Demographic Characteristics, Reduction of Social Interaction, and Perceived Social Support, Worry About Contracting COVID-19, and General Anxiety During the COVID-19 Pandemic (N = 1954).

Variables	*n* (%)	Mean (SD)	Range
Sex			
Women	1305 (66.8)		
Men	649 (33.2)		
Age			
Younger	1000 (51.2)		
Older	954 (48.8)		
Education level			
High (university or above)	1736 (88.8)		
Low (high school or below)	218 (11.2)		
Reduction of social interaction with others			
Family members	217 (11.1)		
Friends	744 (38.1)		
Colleagues or classmates	706 (36.1)		
One form	213 (10.9)		
Two forms	454 (23.2)		
Three forms	182 (9.3)		
Perceived social support			
Family members		3.0 (0.8)	0–4
Friends		2.9 (0.7)	0–4
Colleagues or classmates		2.7 (0.8)	0–4
Worry about contracting COVID-19			
Low	998 (51.1)		
High	956 (48.9)		
General anxiety			
Low	1011 (51.7)		
High	943 (48.3)		

COVID-19: Coronavirus Disease 2019; SD: standard deviation.

**Table 2 ijerph-17-08039-t002:** Association between the Reduction of Social Interaction and Perceived Social Support: Multiple Regression Analysis.

Variables	Perceived Support from Family Members			Perceived Support from Friends			Perceived Support from Colleagues/Classmates		
	B	95% CI of B	*p*	B	95% CI of B	*p*	B	95% CI of B	*p*
Reduction of interactions with ^a^									
Family members	−0.075	−0.300, −0.078	0.001						
Friends				−0.105	−0.220, −0.088	<0.001			
Colleagues/classmates							−0.112	−0.268, −0.115	<0.001
Men ^b^	0.044	0.000, 0.148	0.052	−0.026	−0.107, 0.028	0.243	0.016	−0.049, 0.107	0.479
Older age ^c^	0.074	0.049, 0.191	0.001	−0.002	−0.076, 0.054	0.945	0.076	0.044, 0.194	0.001
Low education level ^d^	−0.050	−0.244, −0.018	0.027	−0.061	−0.240, −0.035	0.007	−0.018	−0.166, 0.070	0.442

CI: Confidence interval. ^a^: No reduction of social contact as the reference; ^b^: Women as the reference; ^c^: Younger age as the reference; ^d^: High education level as the reference.

**Table 3 ijerph-17-08039-t003:** Factors Associated with the Three Types of Reduced Social Interaction: Logistic Regression Analysis.

Variables	Reduction of Interactions with Family Members			Reduction of Interactions with Friends			Reduction of Interactions with Colleagues/Classmates		
	OR	95% CI of OR	*p*	OR	95% CI of OR	*p*	OR	95% CI of OR	*p*
Men ^a^	1.025	0.749, 1.401	0.879	0.993	0.810, 1.219	0.949	1.270	1.036, 1.557	0.022
Older age ^b^	1.388	1.034, 1.863	0.029	1.796	1.477, 2.183	<0.001	1.587	1.305, 1.931	<0.001
Low education level ^c^	0.882	0.543, 1.435	0.614	0.705	0.513, 0.969	0.031	0.749	0.544, 1.030	0.076
High worry about contracting COVID-19 ^d^	2.103	1.536, 2.879	<0.001	2.508	2.051, 3.067	<0.001	2.305	1.884, 2.821	<0.001
High general anxiety ^e^	1.851	1.346, 2.546	<0.001	1.206	0.984, 1.480	0.072	1.210	0.986, 1.485	0.068

CI: Confidence interval; COVID-19: Coronavirus Disease 2019; OR: Odds ratio. ^a^: Women as the reference; ^b^: Younger age as the reference; ^c^: High education level as the reference; ^d^: Low worry as the reference; ^e^: Low general anxiety as the reference.

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
