# Peer review of "Voluntary Reduction of Social Interaction during the COVID-19 Pandemic in Taiwan: Related Factors and Association with Perceived Social Support"

_ijerph, 2020, doi:10.3390/ijerph17218039_

Round 1
Reviewer 1 Report
I would like to congratulate the authors for their interest in researching in this field.
The study has some important limitations which are described in point 4.3 of the paper.
Likewise, the authors should consider other limitations derived from the impossibility of verifying the identity of the individuals included in the survey, as well as the veracity of their responses.
Although this aspect is a representative point of the study, I consider that the topic, redaction and methodological application of the study, compensates for these limitations, resulting in a good final document. That is why I consider that the article can be published with a few minor revisions that I describe below:
- Include in the section on limitations, those derived from the impossibility of verifying the identity of the participants in the survey (or otherwise indicate how this limitation is overcome).
- Include a section on future prospects and possible lines of research based on the data obtained.
- The conclusions should be extended to include a more detailed reflection on the results obtained.
Author Response
We appreciated your comments. As discussed below, we have revised our manuscript with underlines according to your suggestions.
Comment 1
Include in the section on limitations, those derived from the impossibility of verifying the identity of the participants in the survey (or otherwise indicate how this limitation is overcome).
Response
Thank you for your comment. We listed it as one of the limitations in this study as below. Please refer to line 297-299.
“Fourth, participants completed the online questionnaire anonymously. Hence, it is impossible to verify the identity of the respondents and their responses in the survey. It needs to be clarified by a face-to-face interview study with examining the test-retest reliability.”
Comment 2
Include a section on future prospects and possible lines of research based on the data obtained.
Response
Thank you for your comment. We added a paragraph to discuss further research as below. Please refer to line 282-289.
“There are several lines of studies worth further examination based on the data of this study. First, the prediction of personality, coping strategies and psychopathology for voluntary reduction of social interactions during the pandemic warrants study. Second, the follow-up studies are needed to determine the outcomes of the individuals with voluntary reduction of social interactions. Third, what programs that governments and nongovernmental organizations can adopt to increase social support warrants further study. Fourth, cross-cultural comparisons may provide knowledge to understand cultural differences in changes of social relationship during the pandemic.”
Comment 3
The conclusions should be extended to include a more detailed reflection on the results obtained.
Response
Thank you for your comment. In addition to the original contents of Conclusion section, we added new contents as below to reflect the results obtained as below. Please refer to line 307-313.
“The present study identified that age, sex, educational level, worry regarding contracting COVID-19 and general anxiety were significantly associated with all or some aspects of reduced social interactions during the pandemic. The results indicated that the changes in social interaction during the COVID-19 pandemic is a complicated phenomenon involving demographic characteristics and health beliefs. These factors can be used to early identify individuals who reduced social interaction with other people and then perceived low social support.”
Reviewer 2 Report
This short report provides a description of a topic of great current importance. The first wave of COVID-19 infection and the necessity of social distancing has had a major impact on the wellbeing of significant proportions of the population in many countries, and any lessons learned so far can be used to mitigate this as a second wave approaches.
The results were unsurprising, but it is useful to have them confirmed and supported by evidence. The authors acknowledge limitations of the study's approach to recruit participants, but this is counterbalanced by the rapid access to the data allowing concerns about social support to be highlighted in a timely fashion.
I have no suggested amendments to the manuscript.
Author Response
Comment
This short report provides a description of a topic of great current importance. The first wave of COVID-19 infection and the necessity of social distancing has had a major impact on the wellbeing of significant proportions of the population in many countries, and any lessons learned so far can be used to mitigate this as a second wave approaches.
The results were unsurprising, but it is useful to have them confirmed and supported by evidence. The authors acknowledge limitations of the study's approach to recruit participants, but this is counterbalanced by the rapid access to the data allowing concerns about social support to be highlighted in a timely fashion.
I have no suggested amendments to the manuscript.
Response
Thank you for your support!
Reviewer 3 Report
General comments:
I enjoyed reviewing the manuscript of Voluntary Reduction of Social Interaction During the COVID-19 Pandemic in Taiwan: Related Factors and Association with Perceived Social Support”.
The authors started with the discussion of social isolation due to the COVID-19 pandemic and the possible influence on mental health. In general, this paper provided useful information on the influence of reducing social interaction with social support, and the personal anxiety and worry to COVID-19, which could help to keep the public mental health condition during the COVID-19 pandemic.
I recommend the manuscript for publication after some revisions. Below are some detailed comments and suggestions.
- Line 45- 47: “COVID-19 is highly contagious, which has forced governments worldwide to implement policies to reduce the spread of the infection, such as quarantine, lockdown, closure of nonessential stores and educational institutions, and border closure.”.
Relevant references required here. Some web-news or policy document would help to support, or literature like reference 11 and 12 in this paper could be good support.
- Line 133-134: “Responses 1 and 2 and response 3 were classified as voluntarily reducing and not voluntarily reducing social interaction during the COVID-19 pandemic, respectively.”
Is that responses 1 and 2 should be considered as NOT voluntarily reducing social interaction, and response 3 should be considered as voluntarily reducing social interaction?
The sentence needs to be changed as: “Responses 1 and 2 were classified as not voluntarily reducing social interaction during the COVID-19 pandemic, and response 3 were classified as voluntarily reducing social interaction, respectively.”
- Table 1, lines 179-181“Demographic Characteristics, Reduction of Social Interaction, and Perceived Social Support, Worry About Contracting COVID-19, General Anxiety, and Sufficiency of Equipment and Resources Against COVID-19 (N = 1,954).”
In the title, the letter size of “Against COVID-19” need to be reduced to the same as other words.
Also, this table seems did not include some evaluation of “Sufficiency of Equipment and Resources Against COVID-19”. If this is not mentioned here, please remove this part.
Subjects, like “Sex” “Age” ect., need to be highlighted or separated, it is a little bit confusing now.
It might be better to rearrange the title of the table to the same page. Same for line 195, section 3.3
- Table 2
The heading onside need to be rearranged, just add some more space between might help the reader to better get the information.
More space might be required between different columns of data, it a little bit hard to identify the PP value and the B value from the following column.
The CI, confidence interval, and OR, odd ratio, were not presented in this table, please revise or delete them. Better not use the same description with table 3.
The t value and P value both are the evaluation of the significances of individual coefficient in regression, no necessary to show both of them. It might be useful to show some other information like 95% CI.
- Table 3,
Same as table 1 and 2, please consider to rearrange, like add more space between column, and separate 95% CI to an independent line.
- Line 269: “The present study has some limitations warranted discussion.”
Delete the words “warranted discussion”.
Author Response
We appreciated the reviewer’s comments. As discussed below, we have revised our manuscript with underlines according to the reviewer's suggestions.
Comment 1
Line 45- 47: “COVID-19 is highly contagious, which has forced governments worldwide to implement policies to reduce the spread of the infection, such as quarantine, lockdown, closure of nonessential stores and educational institutions, and border closure.” Relevant references required here. Some web-news or policy document would help to support, or literature like reference 11 and 12 in this paper could be good support.
Response
Thank you for your suggestions. We added two references (references 9 and 10) into the revised manuscript to support our descriptions as below. Please refer to lines 46, 48, and 343-349.
Reference 9: Godri Pollitt, K.J.; Peccia, J.; Ko, A.I.; Kaminski, N.; Dela Cruz, C.S.; Nebert, D.W.; Reichardt, J.K.V.; Thompson, D.C.; Vasiliou, V. COVID-19 vulnerability: the potential impact of genetic susceptibility and airborne transmission. Hum Genomics 2020, 14, 17. doi: 10.1186/s40246-020-00267-3.
Reference 10: Chu, D.K.; Akl, E.A.; Duda, S.; Solo, K.; Yaacoub, S.; Schünemann, H.J.; COVID-19 Systematic Urgent Review Group Effort (SURGE) study authors. Physical distancing, face masks, and eye protection to prevent person-to-person transmission of SARS-CoV-2 and COVID-19: a systematic review and meta-analysis. Lancet 2020, 395, 1973-1987. doi: 10.1016/S0140-6736(20)31142-9.
Comment 2
Line 133-134: “Responses 1 and 2 and response 3 were classified as voluntarily reducing and not voluntarily reducing social interaction during the COVID-19 pandemic, respectively.” Is that responses 1 and 2 should be considered as NOT voluntarily reducing social interaction, and response 3 should be considered as voluntarily reducing social interaction? The sentence needs to be changed as: “Responses 1 and 2 were classified as not voluntarily reducing social interaction during the COVID-19 pandemic, and response 3 was classified as voluntarily reducing social interaction, respectively.”
Response
Thank you for your suggestions. We revised this sentence based on your suggestion as below. Please refer to line 144-146.
“Responses 1 and 2 were classified as not voluntarily reducing social interaction during the COVID-19 pandemic, and response 3 was classified as voluntarily reducing social interaction.”
Comment 3
Table 1:
- Lines 179-181“Demographic Characteristics, Reduction of Social Interaction, and Perceived Social Support, Worry About Contracting COVID-19, General Anxiety, and Sufficiency of Equipment and Resources Against COVID-19 (N = 1,954).” In the title, the letter size of “Against COVID-19” need to be reduced to the same as other words.
- This table seems did not include some evaluation of “Sufficiency of Equipment and Resources Against COVID-19”. If this is not mentioned here, please remove this part.
- Subjects, like “Sex” “Age” ect., need to be highlighted or separated, it is a little bit confusing now.
- It might be better to rearrange the title of the table to the same page. Same for line 195, section 3.3
Response
Thank you for your comments.
- We revised “Against COVID-19” into “During the COVID-19 Pandemic” and reduced the letter size be to the same as other words in the revised manuscript. Please refer to line 195.
- We deleted “Sufficiency of Equipment and Resources Against COVID-19” from the title. Please refer to line 194-195.
- We reorganized the table to highlight the subjects in Table 1. Please refer to Table 1.
- We rearranged to make the titles and tables to the same pages. Please refer to Table 1, Table 2 and Table 3.
Comment 4
Table 2
- The heading onside need to be rearranged, just add some more space between might help the reader to better get the information.
- More space might be required between different columns of data, it a little bit hard to identify the PP value and the B value from the following column.
- The CI, confidence interval, and OR, odd ratio, were not presented in this table, please revise or delete them.
- The t value and P value both are the evaluation of the significances of individual coefficient in regression, no necessary to show both of them. It might be useful to show some other information like 95% CI.
Response
Thank you for your comments. We revised the content of Table 2 based on your suggestions as below. Please refer to Table 2.
- We rearranged the headings onside to add more space among them.
- We added more space among different columns of data.
- We deleted the confidence interval and OR from the footnote of Table 2.
- We deleted t value and added 95% CI of B values into Table 2.
Comment 5
Table 3
Same as table 1 and 2, please consider to rearrange, like add more space between column, and separate 95% CI to an independent line.
Response
Thank you for your comments. We revised the content of Table 3 based on your suggestions, including adding more space among different columns of data and separating 95% CI to an independent line. Please refer to Table 3.
Comment 6
Line 269: “The present study has some limitations warranted discussion.”
Delete the words “warranted discussion”.
Response
We deleted “warranted discussion” from this sentence. Please refer to line 291.
Reviewer 4 Report
The study proposes a survey conducted via Facebook to investigate the reduction of social interaction due to the COVID 19 pandemic and the association of reduced social interaction with perceived social support.
The study certainly represents a novelty dealing with the effects of COVID 19 pandemic. However, the manuscript should be improved to solve some criticalities:
- The scientific soundness of the survey should be augmented providing more details on the methodology adopted. In particular, the Authors should explain how data collected were processed and the associations carried out.
- Some graphs illustrating the results could help the reader in better understanding the findings of the survey.
- Language and editing need an improvement.
Author Response
We appreciated the reviewer's comments. As discussed below, we have revised our manuscript with underlines based on the reviewer's suggestions.
Comment 1
The scientific soundness of the survey should be augmented providing more details on the methodology adopted. In particular, the Authors should explain how data collected were processed and the associations carried out.
Response
Thank you for your comment. In the revised manuscript we added introduction for the method we used to collect the data, how data collected were processed, and how the associations were examined as below.
“Briefly, a Facebook advertisement was deployed between April 10 and April 23, 2020. By April 23, 2020, 427 confirmed cases had been identified in Taiwan, of which six were fatal [30]. The inclusion criteria for participants were aged 20 years or older and residing in Taiwan. The Facebook advertisement included a headline, main text, pop-up banner, and link to the research questionnaire website. We designed the advertisement to appear in the Facebook users’ “news feeds,” which is a continually updated list of updates from advertisers and the user’s connections (such as friends and the Facebook groups that they have joined). Our advertisement only targeted users’ news feeds, as opposed to other Facebook advertising locations (e.g. the right column), because news feed advertisements are most effective in recruiting research participants [35]. We targeted the advertisement to Facebook users by location (Taiwan) and language (Chinese), where Facebook’s advertising algorithm determined which users to show our advertisement to. The survey administration software Google Forms was used for data collection (https://www.google.com/forms/about/).” Please refer to line 119-132.
“In total, the data of 1954 respondents (649 men and 1305 women) were analyzed, with 70 of the original 2024 respondents excluded due to missing data.” Please refer to line 184-185.
“Data collected were input, processed and analyzed using the statistical software SPSS (version 22.0; SPSS Inc., Chicago, IL, USA). First, the percentages of respondents who voluntarily reduced the frequency of social interaction with their family members, friends, and colleagues or classmates to avoid COVID-19 infection were calculated. Second, multiple regression analysis was performed to examine the associations of voluntarily reduced social interaction with perceived social support, controlling for the effects of demographic characteristics. Finally, the associations of demographic characteristics, worry regarding contracting COVID-19, and general anxiety with the voluntary reduction of social interaction were examined using multivariate logistic regression with odds ratio and 95% confidence intervals. A two-tailed p value of less than 0.05 was considered statistically significant.” Please refer to line 172-181.
Comment 2
Some graphs illustrating the results could help the reader in better understanding the findings of the survey.
Response
Thank you for your suggestion. In the revised manuscript we added a bar graph to show to various levels of perceived social support in the participants with and without reduction of social interaction. Please refer to Figure 1. Meanwhile, we rearranged the contents of Table 2 and Table 3 to help the readers to better get the information.
Comment 3
Language and editing need an improvement.
Response
Thank you for your comment. We checked the manuscript thoroughly to improve the writing. We invited an English-native editor to correct out manuscript. Some of major revisions were listed below.
- Revise the sentence “Responses 1 and 2 were classified as not voluntarily reducing social interaction during the COVID-19 pandemic, and response 3 was classified as voluntarily reducing social interaction.” Please refer to line 144-146.
- Delete “Sufficiency of Equipment and Resources Against COVID-19” from the title of Table 1. Please refer to line 194-195.
- Delete the confidence interval and OR from the footnote of Table 2.
- Delete “warranted discussion” from the first sentence of Limitation section. Please refer to line 291.
Round 2
Reviewer 3 Report
Great works, no other suggestions.
Reviewer 4 Report
The Authors have augmented the quality of the manuscript sufficiently. Hence. in this reviewer opinion it can be considered for publication.